# Anti-Inflammatory Dietary Diversity and Depressive Symptoms among Older Adults: A Nationwide Cross-Sectional Analysis

**DOI:** 10.3390/nu14235062

**Published:** 2022-11-28

**Authors:** Xiaoyang Lv, Siwei Sun, Jingjing Wang, Huashuai Chen, Shaojie Li, Yang Hu, Mingzhi Yu, Yi Zeng, Xiangyang Gao, Yajun Xu, Yao Yao

**Affiliations:** 1China Center for Health Development Studies, Peking University, Beijing 100191, China; 2Peking University Sixth Hospital, Peking University Institute of Mental Health, NHC Key Laboratory of Mental Health, National Clinical Research Center for Mental Disorders, Peking University Sixth Hospital, Beijing 100191, China; 3Business School, Xiangtan University, Xiangtan 411105, China; 4Center for Healthy Aging and Development Studies, National School of Development, Peking University, Beijing 100871, China; 5Health Management Institute, The Second Medical Center & National Clinical Research Center for Geriatric Diseases, Chinese PLA General Hospital, Beijing 100853, China; 6Department of Nutrition and Food Hygiene, School of Public Health, Peking University, Beijing 100083, China

**Keywords:** dietary diversity, anti-inflammatory diet, depressive symptoms, older adults, China

## Abstract

The purpose of this study was to associate the anti-inflammatory dietary diversity and depressive symptoms among a nationwide sample of Chinese older adults. We used data from the 2018 wave of Chinese longitudinal healthy longevity survey (CLHLS). We assessed depressive symptoms using the 10 items of the center for epidemiologic studies depression scale (CES-D-10). Based on the dietary diversity index (DDI) generated by previous studies, we construct two novel indicators: the protein-enriched dietary diversity index diet (PEDDI) and the anti-inflammatory dietary diversity index diet (AIDDI). We used multivariate logistic models to evaluate the associations of DDI, PEDDI, and AIDDI with depressive symptoms, statistically adjusted for a range of potential confounders. A total of 12,192 participants (mean age 83.6 years) were included in the analysis. We found that participants with a higher score of DDI (OR = 0.91, 95% CI: 0.89–0.92) and PEDDI (OR = 0.91, 95% CI: 0.88–0.93) showed lower odds of having depressive symptoms, while the association between AIDDI and depressive symptoms was more marked (OR = 0.80, 95% CI: 0.78–0.83). The associations remained in subgroup analyses and sensitivity analyses. The results indicate that intaking diversified diet, particularly anti-inflammatory foods, may be associated with a lower risk of depressive symptoms. The findings of this study, if confirmed as causal, provide evidence that an intervention of adopting an anti-inflammatory diversified diet may reduce the burden of depression among older adults.

## 1. Introduction

Depression is a major mental disorder and a leading cause of disability worldwide. Major depressive disorder has been ranked as the third cause of the burden of disease worldwide in 2008 [1]. In China, depressive disorder ranks second in the list of causes of disability [2]. Depression is common throughout life, with up to one-third of older adults experiencing depressive symptoms [3]. In the context of an aging population, depressive disorder has become a major mental illness for older adults in China; its health service utilization rate is low, and the rate of adequate treatment is very low. A growing body of studies have focused on risk factors for depression, such as diet, in the hope of promoting healthy longevity and quality of life in humans.

Earlier studies have found that many dietary patterns are associated with improved mental health [4]. A Mediterranean-style dietary pattern [5,6] and a healthy diet pattern (a high intake of fruits, vegetables, fish, and whole grains) [7] were both associated with a lower prevalence of depressive symptoms. A DASH diet [8] and a whole-food pattern [9] were related to lower depression risk. A recent study comparing pro-inflammatory and anti-inflammatory diets showed a significant association between a pro-inflammatory diet and an increased risk of a depression diagnosis or depressive symptoms [10].

In studies of older populations, dietary diversity has been found to be linked to health outcomes such as cognitive function [11] and psychological resilience [12] in older adults. Despite the fact that the older adults are a high-risk group for mental health problems, studies conducted on dietary patterns and depressive symptoms among older Chinese are still scarce. Previous studies on dietary structure were mainly carried out in developed countries, and there is less evidence from developing countries. Moreover, previous studies have reported the influence of Mediterranean diet, a healthy and traditional diet, the DASH diet, and dietary diversity on mental health but did not further divide the influence of protein diversity diet and anti-inflammatory diet on mental health, including depression.

In this study, we assessed depressive symptoms with structured questionnaires for participants by using data from the CLHLS cohort. We constructed a dietary diversity index (DDI) by summing up the frequency of intaking 9 types of foods commonly consumed in China and considered two dietary patterns: a protein-enriched diet and an anti-inflammatory diet. Our aim was to study the association of dietary diversity with depressive symptoms in the old population aged 65 years or older.

## 2. Materials and Methods

### 2.1. Study Population

Data used in this study were derived from the Chinese longitudinal healthy longevity survey (CLHLS) waves of 2018. A multistage, stratified cluster sampling design was used by the CLHLS in 23 of China’s 31 provinces. The specifics have been discussed [13]. In 2018, CLHLS used the CED-S to investigate depression for the first time. Therefore, this study used the data of the 2018 wave to investigate the association between the DDI and depressive symptoms. After eliminating 3682 individuals with missing data on important covariables, the final analytical sample utilized in this study contained 12,192 adults aged 65 years or older. All participants and/or their family provided informed permission, and the study was authorized by Peking University’s Ethics Committee (IRB00001052-13074).

### 2.2. Assessment of Dietary Diversity 

The dietary diversity index (DDI) was measured at baseline using a food-frequency questionnaire relating to the nine primary food groups: including vegetables, fruits, legumes and their products, nuts, meat, eggs, fish, dairy and its products, tea, and cereals. DD was constructed using these 9 food groups. Consumption of any food category “frequently or virtually every day” was deemed to equal one DDI unit and DDI score, ranging from 0 to 9 [14]. Cereals and oil were not included in the construction of the DDI since almost all Chinese consume these two food groups daily. We considered six sources of protein-enriched foods: meats, fish, eggs, dairy and its products, nuts, and bean products. We constructed the protein-enriched diet diversity index (PEDDI), which ranged from 0 to 5, by summing up the frequency of intake of protein-rich food. The details of the protein-enriched foods have been described in an earlier study [15]. Based on previous research, the anti-inflammatory diet diversity index (AIDDI) was estimated with reference to the dietary inflammatory index (DII), ranging from 0 to 5. The DII scoring algorithm is based on primary research articles published through December 2010 on the effect of diet on inflammation [16]. It evaluates a total of forty-five food parameters based on the effects of each dietary parameter on six inflammatory biomarkers (interleukin-1β, interleukin-4, interleukin-6, interleukin-10, tumor necrosis factor-α, and C-reactive protein). In our study, the anti-inflammatory food includes vegetables, fruits, legumes and their products, nuts, and tea. Participants’ food histories were questioned to determine daily dietary grams, which indicate weekly average intake. The consumption of any food category above “frequently or virtually every day” was deemed to equal one AIDDI unit. The DII score methodology is based on major research on the impact of diet on inflammation.

### 2.3. Assessment of Depressive Symptoms

We used the 10 items of the center for epidemiologic studies depression scale (CES-D-10), ranging from 0 to 30, to measure depressive symptoms in this study [17]. A higher CES-D-10 score was associated with more severe depressive symptoms. Those with a score of at least 10 are considered to have depressive symptoms [18], which has been well verified in several studies on depressive symptoms measurement in Chinese older population [19] (Cronbach’s alpha is 0.807).

### 2.4. Covariates

To minimize the effect of potential confounders, we controlled for a large number of potential confounding factors [18], which included age, sex (male or female), Hukou (urban or rural), ethic status (“Han” or “others”), and education (0, 1–6, and ≥7 years). Smoking and drinking status were dichotomous variables (Yes or No). The body mass index (BMI) was calculated as weight in kilograms divided by height in meters squared. The Chinese version of the 30-point mini-mental state examination (MMSE) was used to test cognitive function with a cut-off of 24 [20]. The Katz index was used to evaluate the index of activity of daily living (ADL) [21], and ADL disability was defined as incontinence or the need for help in executing one or more of the five fundamental activities [22]. Among them, if the participant needs personal assistance or incontinence in eating, we define “With difficulty in self-feeding = yes”, otherwise “With difficulty in self-feeding = no”. Based on the participants’ replies, physical activities were classified as “yes” or “no”.

### 2.5. Statistical Analysis

Multiple imputation methods were performed to fill in missing covariates. Subject characteristics were compared using an analysis of variance or chi-square test, as appropriate. We used multivariate logistic models to evaluate the associations of depressive symptoms with DDI, PEDDI, and AIDDI, respectively. The base model (Model 1) controlled for gender and age; Model 2 further controlled for hukou, ethnic, BMI, and education; Model 3 additionally controlled for physical activities, smoking, and drinking; and Model 4 added difficulty in self-feeding, dementia, and tooth count in Model 3.

We conducted subgroup analyses to examine whether the associations of depressive symptoms with dietary diversity (DDI, PEDDI, and AIDDI) differed by gender, age (< 75, 75–89 and ≥90), Hukou (urban or rural), and smoking and drinking status (Yes or No), with difficulty in self-feeding or not. We performed several steps of sensitivity analyses for the full model (Model 4). First, we excluded the participants with severe cognitive impairment with scores of MMSE < 21 due to concerns that they may have recall biases in reporting food frequency. Second, older adults who were long bedridden or terminally ill were excluded for more robust estimates. Moreover, we tested our results using the sample prior to multiple imputations. SAS 9.4 statistical software was used to perform all statistical analyses.

## 3. Results

### 3.1. The Characteristics of Study Participants

Our study used information on 12,192 included participants. The mean of the dietary diversity index of them was 4.1 ± 2.1. The mean age of the participants was 83.6 ± 11.1 years; the composition of male and female was relatively balanced (46.5% men). The participants were almost all Han nationality (94.9%) and most of their hukou were in rural (71.3%). The overall education level of the participants was low, with only 26.6% of participants having received more than 7 years of education; 16.1% smoked and 15.3% drank alcohol, 33.8% had physically activities, and 18.3% had a disability in activities of daily living (ADL) (Table 1). Compared to the participants with the lowest quartile of DDI, the participants who had the higher quartile of DDI as a whole in the 2018 CLHLS sample were more likely to be men, have lived in urban rather than rural areas, had a higher educational status, and have been more active in physical activities. However, the participants who had the higher quartile of DDI were more likely to be alcohol drinkers as well.

### 3.2. DDI and Depressive Symptoms

The participants who had the higher score of dietary diversity (DDI, PEDDI, and AIDDI) showed lower odds of having depressive symptoms in model 1 controlling for demographic variables: dietary diversity index (DDI), OR = 0.88 (95% CI: 0.87–0.90); protein-enriched diet diversity index (PEDDI), OR = 0.87 (95% CI: 0.85–0.90); and anti-inflammatory diet diversity index (AIDDI), OR = 0.77 (95% CI: 0.74–0.79). After additional socioeconomic status and health characteristics were controlled for in model 2 and model 3, the magnitude of the association was lowered but remained statistically significant. In the final model, which took into account all of the confounding risk variables, dietary diversity was still linked to decreased odds of experiencing depressive symptoms by 9% (DDI), 9% (PEDDI), and 20% (AIDDI), respectively (Table 2).

We discovered that the associations were robust across the following subgroups of depressive symptoms risk variables based on subgroup analysis, such as gender, age, Hukou, and smoking and drinking status. We observed a statistically significant interaction of smoking status with all three dietary diversity categories (DDI, PEDDI, and AIDDI) on depression symptoms. The association between dietary diversity categories (DDI, PEDDI, and AIDDI) and depressive symptoms was more marked in smokers (Table 3).

### 3.3. Sensitivity Analyses

We obtained similar results using the full model (Model 4) through a sensitivity analysis of the different included populations. All of the relationships between dietary diversity categories (DDI, PEDDI, and AIDDI) and depression symptoms were consistent with the final model in the main article, after we (1) excluded the participants with severe cognitive impairment with scores of MMSE < 21, (2) excluded the older adults who were long bedridden or terminally ill, and (3) used the sample prior to multiple imputations (Appendix A).

## 4. Discussion

We found the following main findings in this large population-based study. First, Chinese older adults with higher scores of dietary diversity (DDI, PEDDI, and AIDDI) were less likely to show depressive symptoms. There is a strong association between anti-inflammatory dietary patterns and depressive symptoms. The associations remained in subgroup analyses and sensitivity analyses.

Our findings support evidence from several past studies in various age groups that diet has a great influence on mental health. J Rienks et al. found that consumption of a ‘Mediterranean-style’ dietary pattern by mid-aged women may have a protective influence against the onset of depressive symptoms [5]. O’Neil A et al. discovered evidence of a strong, cross-sectional link between poor mental health and an unhealthy diet in both children and adolescents [23]. For women aged 20–49, a one-point increase in the dietary diversity score was related to a 39% lower odds of depression [24]. Dietary diversity research has effectively avoided the restrictions imposed by specific nutrients and foods. Our past research has observed the positive effects of several nutrient combinations on the cognitive health of older adults in China, which supported the research on young people [11]. Many studies have confirmed that dietary diversity has a protective effect on depression, which may be related to the high proportion of ingredients in food groups such as milk and dairy products, vitamin-rich fruits and vegetables, eggs, beans, and nuts [25,26]. A research of Chinese aged persons found a negative correlation between vitamin D levels and depressive symptoms [25]. Tyrosine hydroxylase gene expression is regulated by vitamin D, which is also essential for the production of norepinephrine and dopamine [26]. Coenzymes generated from vitamins B6, B12, and folate are essential for the production and metabolism of serotonin and dopamine. The inadequate dietary intake of these vitamins may result in homocysteine buildup and decreased monoamine synthesis in the brain, which may be a key factor in the genesis of depression [24]. Our results of protein-enriched diet are consistent with a study that observed that high protein intake was significantly related to a reduced prevalence of depression after classifying participants according to protein dietary patterns [27]. Other studies also have found that a high-protein diet including egg and milk may have a greater beneficial effect on mental health. Tryptophan and serotonin levels have been researched in relation to how protein consumption affects behavior and mood. In fact, tryptophan’s conversion to serotonin gives it an impact similar to an antidepressant [28]. 

Many studies of inflammatory diet and depression support our claim of an anti-inflammatory diet. A pro-inflammatory diet may be linked with a higher incidence of depressive symptoms in a cohort of older Americans and those in the most pro-inflammatory group had a roughly 24% greater chance of developing depressive symptoms when compared to those with the most anti-inflammatory diet [29]. Shivappa N et al. suggested that an anti-inflammatory diet (RR = 0.81; 95% CI: 0.69–0.96; P_trend = 0.03) is associated with a lower risk of depression in middle-aged Australian women. Women with the most anti-inflammatory diet had an approximately 20% lower risk of developing depression than women with the most pro-inflammatory diet [30]. Chronic low-grade inflammation; cell-mediated immunity; and the compensatory anti-inflammatory reflex system (CIRS), which is characterized by inverse immunoregulatory processes, are all linked to depression [31]. A pro-inflammatory diet was linked to a significantly increased risk of depression. The HR for individuals in the highest percentile of the dietary inflammatory index (highly pro-inflammatory) was 1.47 (95% CI: 1.17–1.85), with a significant dose-response association (= 0.01) [32]. According to Rienks J, following the Mediterranean dietary pattern, which includes garlic, a frequent anti-inflammatory food, is protective against depression [5]. Garlic alleviates depression-related behaviors possibly by attenuation of brain oxidative stress [33].

The correlations between the three dietary varieties and depressive symptoms were intact after subgroup analysis. Most studies are consistent with our findings that dietary diversity improves depression in both men and women [5,9]. It is also worth mentioning that the associations were more marked in smokers in our study. Having mental illness and smoking tend to co-occur often, which is a serious public health risk [34]. In a study of middle-aged and older adults, smokers had a 20% (95% CI: 12–28%) higher risk of developing depression throughout the follow-up period [35], supporting the theory that persistent smoking increases susceptibility to depression [36]. Our research discovered that the physical advantages of anti-inflammatory diet diversity were more prominent in smokers, implying that greater dietary diversity may partially counterbalance the mental health risks of smoking. Oxidative stress, inflammation, and atherosclerosis processes are potential causes of smoke’s negative effects [37]. We indicated in the previous paragraph that an anti-inflammatory diet might help ease depression-related symptoms by lowering oxidative stress, so it may help mitigate some of the harmful consequences of smoking.

Some strengths of the study include a substantial sample size, a lengthy period of follow-up, and adjustment for multiple potential confounders. In addition, we used the center for epidemiologic studies depression scale (CES-D-10) and the food-frequency questionnaire to assess depressive status and eating habits in large population studies. The depression scale is well-known for being a simple and effective test for measuring depression.

Regarding the research design, there were a number of restrictions. First of all, because dietary data are self-reported and could not accurately reflect real food intake or objective depressive condition, we cannot completely exclude memory bias. Secondly, the dietary diversity in this study is frequency, which is semi-quantitative, and lacks the relationship between quantitative dietary diversity and health outcomes. More detailed research design is required to study related issues. Thirdly, the outcome indicators belong to epidemiological investigations, and there may be measurement bias. However, we applied different cutoff values in the sensitivity analysis and found that the results were robust, which maximized the avoidance of this bias.

In conclusion, this population-based study suggests that diversified diet intake, particularly anti-inflammatory diets, is related to a lower prevalence of depressive symptoms among older adults. Specifically, this study highlights the significance of adopting a diversified diet, with a special emphasis on promoting the consumption of anti-inflammatory diets, which can decrease the incidence of depression in older persons. More longitudinal research is required to confirm our findings to provide evidence that an intervention of adopting an anti-inflammatory diversified diet may reduce the burden of depression among older adults.

## Figures and Tables

**Table 1 nutrients-14-05062-t001:** Baseline characteristics of the study sample by dietary diversity index (DDI) quartiles.

Variable/Subgroups	Total Sample *N* = 12192	Q1 *N* = 3048	Q2 *N* = 3048	Q3 *N* = 3048	Q4 *N* = 3048	*p* Value
Dietary Diversity Index, score	4.1 ± 2.1	1.4 ± 0.7	3.4 ± 0.5	4.7 ± 0.5	6.9 ± 1.0	<0.001
Age, years	83.6 ± 11.1	84.0 ± 11.0	83.6 ± 11.1	83.7 ± 11.0	83.1 ± 11.1	0.02
BMI, kg/m [2]	22.5 ± 3.9	22.1 ± 4.0	22.6 ± 4.0	22.4 ± 3.9	22.9 ± 3.8	<0.001
gender, %						
Men	5666 (46.5%)	1204 (39.5%)	1397 (45.7%)	1467 (48.2%)	1598 (52.4%)	<0.001
Women	6526 (53.5%)	1843 (60.5%)	1659 (54.3%)	1575 (51.8%)	1449 (47.6%)	
Hukou, %						
Urban	3499 (28.7%)	385 (12.6%)	675 (22.1%)	898 (29.5%)	1541 (50.6%)	<0.001
Rural	8693 (71.3%)	2662 (87.4%)	2381 (77.9%)	2144 (70.5%)	1506 (49.4%)	
Ethic, %						
Han	11567 (94.9%)	2795 (91.7%)	2917 (95.5%)	2894 (95.1%)	2961 (97.2%)	<0.001
Others	625 (5.1%)	252 (8.3%)	139 (4.5%)	148 (4.9%)	86 (2.8%)	
Education, %						
0	6481 (53.2%)	2012 (66.0%)	1735 (56.8%)	1575 (51.8%)	1159 (38.0%)	<0.001
1–6	2472 (20.3%)	587 (19.3%)	643 (21.0%)	662 (21.8%)	580 (19.0%)	
>6	3239 (26.6%)	448 (14.7%)	678 (22.2%)	805 (26.5%)	1308 (42.9%)	
ADL, %						
Independency	9965 (81.7%)	2561 (84.0%)	2459 (80.5%)	2488 (81.8%)	2457 (80.6%)	<0.001
Dependency	2227 (18.3%)	486 (16.0%)	597 (19.5%)	554 (18.2%)	590 (19.4%)	
Physical activities, %						
No	8075 (66.2%)	2307 (75.7%)	2121 (69.4%)	1968 (64.7%)	1679 (55.1%)	<0.001
Yes	4117 (33.8%)	740 (24.3%)	935 (30.6%)	1074 (35.3%)	1368 (44.9%)	
Smoking, %						
No smoking	10227 (83.9%)	2583 (84.8%)	2558 (83.7%)	2531 (83.2%)	2555 (83.9%)	0.41
Smoking	1965 (16.1%)	464 (15.2%)	498 (16.3%)	511 (16.8%)	492 (16.1%)	
Drinking, %						
No drinking	10321 (84.7%)	2672 (87.7%)	2607 (85.3%)	2544 (83.6%)	2498 (82.0%)	<0.001
Drinking	1871 (15.3%)	375 (12.3%)	449 (14.7%)	498 (16.4%)	549 (18.0%)	

**Table 2 nutrients-14-05062-t002:** Associations of types of Dietary Diversity, Dietary Pattern with depressive symptoms among whole sample.

	Diet Diversity Index (DDI)	Protein–Enriched Diet Diversity Index (PEDDI)	Anti–Inflammatory Diet Diversity Index (AIDDI)
**All–cause Mortality**			
Model 1	0.88 (0.87–0.90) *	0.88 (0.85–0.90) *	0.77 (0.74–0.79) *
Model 2	0.90 (0.88–0.92) *	0.90 (0.88–0.93) *	0.79 (0.76–0.81) *
Model 3	0.91 (0.89–0.93) *	0.91 (0.89–0.94) *	0.81 (0.78–0.83) *
Model 4	0.91 (0.89–0.93) *	0.91 (0.88–0.93) *	0.80 (0.78–0.83) *

* *p* < 0.05.

**Table 3 nutrients-14-05062-t003:** Associations of types of Dietary Diversity, Dietary Pattern with depressive symptoms among subpopulations.

Variable/Subgroups	Diet Diversity Index (DDI)	P–Interaction	Protein–Enriched Diet Diversity Index (PEDDI)	P–Interaction	Anti–Inflammatory Diet Diversity Index (AIDDI)	P–Interaction
**gender**						
Men	0.90 (0.87–0.92) *	0.14	0.90 (0.86–0.94) *	0.30	0.79 (0.75–0.83) *	0.12
Women	0.92 (0.89–0.94) *		0.92 (0.88–0.95) *		0.82 (0.78–0.86) *	
**Age**						
<75	0.91 (0.88–0.95) *	0.76	0.94 (0.89–1.00) *	0.34	0.77 (0.72–0.82) *	0.13
75–89	0.89 (0.87–0.92) *		0.88 (0.84–0.92) *		0.79 (0.75–0.83) *	
>=90	0.92 (0.89–0.95) *		0.91 (0.87–0.96) *		0.85 (0.80–0.90) *	
**Hukou**						
Urban	0.92 (0.89–0.95) *	0.18	0.95 (0.90–1.01) *	0.01	0.80 (0.76–0.85) *	0.36
Rural	0.90 (0.88–0.92) *		0.89 (0.86–0.92) *		0.80 (0.77–0.83) *	
**Smoking**						
No smoking	0.92 (0.90–0.94) *	0.03	0.92 (0.89–0.95) *	0.04	0.81 (0.78–0.84) *	0.04
Smoking	0.86 (0.82–0.91) *		0.85 (0.79–0.91) *		0.74 (0.68–0.81) *	
**Drinking**		0.20		0.33		0.14
No drinking	0.91 (0.89–0.93) *		0.91 (0.89–0.94) *		0.81 (0.78–0.84) *	
Drinking	0.87 (0.83–0.92) *		0.86 (0.80–0.93) *		0.76 (0.70–0.83) *	
**With difficulty in self-feeding**						
Yes	0.94 (0.83–1.06)	0.56	0.90 (0.75–1.07)	0.94	0.95 (0.78–1.15)	0.06
No	0.91 (0.88–0.93) *		0.91 (0.88–0.93) *		0.80 (0.77–0.83) *	

Model 4 controlling for gender, age, hukou, ethnic, BMI, education, physical activities, smoking, drinking, self-feeding in ADL disability, dementia, tooth count. * *p* < 0.05.

## Data Availability

The data of CLHLS are available at https://opendata.pku.edu.cn/dataverse/CHADS (accessed on 28 September 2022).

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
