# Peer review of "Anti-Inflammatory Dietary Diversity and Depressive Symptoms among Older Adults: A Nationwide Cross-Sectional Analysis"

_nutrients, 2022, doi:10.3390/nu14235062_

Round 1
Reviewer 1 Report
In this large population-based study, the Authors indicate that intaking diversified diet, particularly anti-inflammatory foods, may be associated with a lower risk of depression. For this purpose the following are evaluated:1) dietary diversity index (DDI) 2) protein-enriched diet diversity index (PEDDI) 3) anti-inflammatory diet diversity index (AIDDI) 4) Center for Epidemiologic Studies Depression Scale to measure depressive symptoms. Therefore it is observed that Chinese older adults who had the higher score of dietary diversity (DDI, PEDDI and AIDDI) were less likely to show depressive symptoms.
Methods: Insert in the text a brief note of how the anti-inflammatory diet diversity index (AIDDI) is evaluated. Is a drug history also included in the evaluation of depressive symptoms?
Results: In the text (lines 177-180) it says “Compared to the participants with the lowest quartile of DDI, the participants who had the higher quartile of DDI … had a higher educational status and more active in physical activities”. However in table 1, the percentage of those who do not exercise, in the top quartile (55.1%), is higher than those who exercise (44.9%).
line 206: cancel "the participants"
Discussion: Is the statement "In particular, the anti-inflammatory dietary patterns are more strongly associated with depressive symptoms" correct? (lines 212-214)
Author Response
In this large population-based study, the Authors indicate that intaking diversified diet, particularly anti-inflammatory foods, may be associated with a lower risk of depression. For this purpose the following are evaluated:1) dietary diversity index (DDI) 2) protein-enriched diet diversity index (PEDDI) 3) anti-inflammatory diet diversity index (AIDDI) 4) Center for Epidemiologic Studies Depression Scale to measure depressive symptoms. Therefore, it is observed that Chinese older adults who had the higher score of dietary diversity (DDI, PEDDI and AIDDI) were less likely to show depressive symptoms.
Methods: Insert in the text a brief note of how the anti-inflammatory diet diversity index (AIDDI) is evaluated. Is a drug history also included in the evaluation of depressive symptoms?
Response: Thanks for your valuable suggestions, and we have added a brief explanation of how to evaluate the anti-inflammatory diet diversity index (AIDDI) in section 2.2.part (lines 141-151). Additionally, a drug history is not included in the evaluation of depressive symptoms. In accordance with previous literature, we controlled for a number of potential confounding factors. Considering physical health, we selected two of the most used factors: mental state and activity ability, which enabled our model to better control the influence of potential confounding factors.
Results: In the text (lines 177-180) it says “Compared to the participants with the lowest quartile of DDI, the participants who had the higher quartile of DDI … had a higher educational status and more active in physical activities”. However in table 1, the percentage of those who do not exercise, in the top quartile (55.1%), is higher than those who exercise (44.9%).
Response: We thank for the useful suggestions. The meaning of this sentence is that in the group with higher DDI scores, a larger proportion of individuals exercised and a smaller percentage did not. Specifically, the proportion of people who exercised in the top quartile of the DDI population was very small, only 24.3%. By contrast, in the other three groups, the proportion of exercisers gradually increased (30.6%, 35.3%, 44.9%, respectively), consistent with the increasing trend in the DDI scores.
line 206: cancel "the participants"
Response: Thanks for the reminder, we have removed "the participants" in the corresponding position.
Discussion: Is the statement "In particular, the anti-inflammatory dietary patterns are more strongly associated with depressive symptoms" correct? (lines 212-214)
Response: Thanks for the comment. According to the analysis results, the adjusted OR value of AIDDI is always greater than DDI, so it is believed that anti-inflammatory diets are more likely to reduce depressive symptoms. "A varied diet that focused on reducing inflammation had a greater impact on reducing depressive symptoms than a general varied diet." (lines 279-280)

Reviewer 2 Report
Interesting study with detailed information relevant to depression patients.
Author need to include the demographic information and other know variables (Age, sex, known drug use etc.).
Language needs to be checked carefully.
Author Response
Review 2
Interesting study with detailed information relevant to depression patients.
Author need to include the demographic information and other know variables (Age, sex, known drug use etc.).
Response: We appreciate your helpful comment. We agree with your term that the demographic information and other know variables should be included. We showed age, sex,Hukou, ethic, educational level, etc. by DDI groups in Table 1. The proportion of antidepression drug use is very low among older population. Our results may largely be robust if we further consider this. We acknowledge this in limitation section.
Language needs to be checked carefully.
Response: We thank for the thoughtful comments. We have edited the manuscript in revised version.
